# No Media, No Voters? The Relationship between News Deserts and Voting Abstention

**Giovanni Ramos** *, **Luísa Torre and Pedro Jerónimo**

LabCom—Communication and Arts, Faculty of Arts and Letters, University of Beira Interior,
6201-001 Covilhã, Portugal; luisa.torre@ubi.pt (L.T.); pj@ubi.pt (P.J.)
* Correspondence: gio@gioramos.net

**Abstract:** Local journalism has suffered major transformations as traditional business models collapse and habits of news consumption change. A lack of funding and successive economic crises have brought about, on a global scale, the shutdown of many news outlets in smaller territories. These areas are becoming "news deserts", a phenomenon that has been mapped in Brazil and Portugal. Territories without news could see an uptick in social problems such as disinformation, populism, and democratic crises, especially because of voting abstention. Background: This paper aims to analyze the relationship between news deserts and democracy, focusing on how news deserts correlate with voting abstention rates in Brazil and Portugal. Methods: A literature review was carried out including data from news deserts in both countries. The abstention rates in this analysis concern national elections held in 2022. A correlation analysis using binary logistic regression was deployed comparing municipalities with the highest and the lowest abstention rates. Results: In both countries, it was not possible to assess whether there was a correlation between abstention rates and the existence of news deserts. Conclusions: While the absence of media outlets is not correlated with the mobilization of citizens to vote, other variables may be affecting voters' abstention behaviors.

**Keywords:** news deserts; journalism crisis; local journalism; democracy; abstention rates





## 1. Introduction

When the first newspapers migrated to the internet in the 1990s, new business possibilities opened up for media companies beyond the traditional print business, which had been experiencing a readership drop since the second half of the 20th century (Anderson et al. 2012; Anderson 2014; Costa 2014). However, newspapers, radio and TV stations struggled to make the transition to digital and address the revenue decline. In Portugal, the printed press's revenue dropped 75% in 10 years (M. Botelho 2017), while in Brazil, it more than halved between 2007 and 2018 (Accenture 2021).

For journalism, the ICT revolution has impacted the media business model, resulting in the loss of financial income and audiences to new models linked to the internet. Advertising, the main source of revenue for newspapers and radio and TV stations (Anderson et al. 2012), started to migrate from traditional media to large digital platforms such as Google, Facebook or Amazon, which control 70% of digital ads published in the United States (Hindman 2018). Companies that prefer to advertise on social media and other platforms, to the detriment of newspapers and native digital journalism outlets, do so because they feel it is easier to reach their target audience (Hindman 2015).

The loss of advertisements to digital platforms made newspapers rely more on circulation to obtain revenue. The biggest difficulty was in changing consumer habits. Audiences prefer to obtain their information on mobile devices such as smartphones and tablets and do not want to pay for the news (Quintanilha 2018). The digital consumer still has difficulties identifying access to news over the internet as a product to be paid for (Christofoletti 2019). In Brazil, only 18% of respondents showed interest in paying for online news, while in Portugal, this figure is even lower at 12% (Newman et al. 2022).

The problems faced by journalism after the ICT revolution are aggravated in local journalism, especially far from large metropolises. Small newspapers, local radio and regional TV stations naturally face challenges for sustainability due to their geographic limitation that hinders the growth of their audience (Hindman 2018).

In most cases, local journalism's proximity to their audiences narrows the newspaper's coverage area and limits its scalability, either because the municipality it reports on is not very populated or because it is surrounded by large metropolises. In addition, local newspapers remain attached to the pre-internet business model: they see the digital world as a threat to their business, often disregarding all the potential that online journalism could bring them (Jerónimo 2015).

In addition to the problems of expanding their audiences, local media outlets face competition from other media and other services when vying for internet audiences. Data from comScore in the United States indicate that only 3% of visits to pages on the internet are for news sites. Additionally, within this universe, only one sixth are for local news (Hindman 2018). That is, only half a percent of visits to pages on the internet in the US are on local newspaper, radio or TV sites.

Camponez (2017) points out another challenge for local newspapers: competition from other platforms in areas where local journalism has always had an advantage, such as providing information about services. Today, when someone is looking for a service in their city, the local newspaper competes with search engines such as Google.

With the fall in investment in print newspapers, local media need to find new ways of performing in the market. Olsen and Solvoll (2018) claim that the local press needs to look for sources of digital revenue if they want to continue to act as mediators of the local debate. In this sense, "building a sustainable digital revenue model also has wider societal implications" (Olsen and Solvoll 2018, p. 174).

## 2. Literature Review

### 2.1. News Deserts, Disinformation and Democracy

News deserts, as a phenomenon, are the reflection in local media of the crisis of journalism in the 21st century. The concept refers to the existence of communities where there is no local news production, or a deficient level of production, below the needs of the community. When defining news deserts, it is important to note that they are not just communities without their own newspapers, but also with difficulty in accessing local news.

We defined a "news desert" as a community without a local newspaper. As a result of the dramatic shrinkage in the number of local news outlets in recent years, as well as the decrease in local news coverage by surviving newspapers, we have expanded our designation of news deserts to include communities where residents are facing significantly diminished access to the sort of important news and information that feeds grassroots democracy (Abernathy 2018).

The United States was the first country to map its news deserts, through the "Expanding News Deserts" project. Data show that the phenomenon mainly affects poorer, elderly and less educated communities in the United States. This is because newspaper closures were more severe in more isolated cities, far from larger metropolises, in which there is a smaller population and less economic activity (Abernathy 2018, 2020).

In addition to the issues brought forth by smaller populations and lower economic activity, local newspapers in the inland regions of countries have another problem: their audience tends to be older, a fact that hinders the digital transition of these newspapers. However, when they do transition to digital, they face another challenge: programmatic advertising, the most common form of monetization on the internet, relies on large audiences to work, something that a local newspaper can struggle to reach (Hindman 2015; Jerónimo et al. 2022).

In Caroline County, Virginia, one of 225 places in the United States that has become a news wasteland (Abernathy 2018), Mathews (2022) pointed to a weakening of county

residents' sense of community following the closure of the single newspaper, which was seen by residents to unite the entire county into a single society.

They felt Caroline County was often placed in a negative light by outside media publications, but the community weekly newspaper made them feel "special" and proud of their county. Especially through event notifications, the newspaper brought county residents together. Without the newspaper, participants said they were isolated to smaller neighborhoods and missed the connection to a larger county-wide community (Mathews 2022, p. 1261).

The emergence of news deserts is particularly challenging at a time when misinformation and disinformation in local contexts spread fast through digital media (Jerónimo and Esparza 2022). The study "Local News Deserts in the UK" showed that, as local newspapers shut their doors, attention to local events turned online. Social media became central in local news distribution systems, both for information produced by communities within social groups as well as for the distribution of news produced by local media. In this context, social media are seen as a main source of local disinformation, due to the lack of fact-checking in these networks (Barclay et al. 2022; Correia et al. 2019).

Local media have been facing an unprecedented crisis, but local news continues to be an essential tool for civic engagement. "Without news media providing this civic function, the public becomes less informed about issues that affect them and there is an agenda-setting vacuum left behind" (Marwick and Lewis 2017, p. 42) Thus, social groups, leveraging techniques of participatory culture and the affordances of social media, manage to influence the public sphere by spreading their beliefs and disseminating disinformation to the detriment of content produced and checked by journalists (Marwick and Lewis 2017).

In a local context, where local media struggle with a confluence of economic and technological factors, communities that become news deserts face an agenda-setting vacuum, where the citizens have access to trustworthy information about what is going on in their countries or in the world, but not about their local communities. People that live in these news deserts, however, still receive their local news. Now, it comes mostly from social media, where verification is not a regular practice, and the spread of disinformation becomes an imminent threat (Barclay et al. 2022; Jerónimo and Esparza 2022). Additionally, although the public pays attention to news published on Facebook and Twitter by traditional media and journalists, these reliable sources are increasingly being overshadowed by influencers and other alternative sources, along with a long-term trend of growing disinterest in the news itself (Abernathy 2022; Jerónimo and Esparza 2022; Marwick and Lewis 2017). As a result, those who do not trust the media are more likely to have lower levels of political knowledge, which may impede their full exercise of their democratic rights.

The solution to this situation "goes to deeper issues of repairing political institutions and democratic values" (Bennett and Livingston 2018, p. 124). We argue that empowering and supporting local media are also impactful ways to fight the disinformation phenomenon (Jerónimo and Esparza 2022).

Many studies examine the relationship between the existence of local news and their civic participation. The existence of newspapers by the beginning of the 20th century was assessed to have a robust positive effect on political participation (Gentzkow et al. 2011). The literature also shows that the closure of a local newspaper from a community can lead, for example, to less civic engagement (Shaker 2014) or lower voter turnout (Schulhofer-Wohl and Garrido 2009), while the offering of politically relevant information by local media increases voting turnout (Baekgaard et al. 2014). However, in a changing news landscape, more recent studies show that turnout rates in major party elections do not significantly covary with an absence of local papers; that is, local news circulation is unrelated to higher turnout (Chapp and Aehl 2021).

Despite being frequent in the literature, the idea that local journalism plays an important role in strengthening democracy and citizenship and could be a central actor in combating disinformation is questioned by Usher (2023). She cites a study carried out in the 1990s by Eliasoph that points to problems in local political coverage by local newspapers in

the United States. It shows that, in addition to low political coverage, these newspapers had a bias that would compromise the idea of journalism as a public arena.

Usher (2023) also relates the concept of news deserts to the fight against disinformation, as the news desert framing carried out in the United States only includes legacy media, ignoring the presence of journalism in other nontraditional formats that can supply the demand for information in a community. Another note by Usher (2023) is the framing by municipalities or counties, in the case of the US, which does not shed light on inequalities within the territories. An example is the city of Chicago, which has local media outlets, but they are all geared towards the richest part of the city. That is, Chicago's poor neighborhood, a community, is also neglected by local information, but the municipality falls outside the news desert.

Assessing if there is a correlation between absence rates in elections and the (in)exist ence of local media can be a useful way to start addressing to what extent local media play a relevant role in strengthening democracy at a local level.

### 2.2. News Deserts Scenarios in Brazil and Portugal

The mapping of news deserts has been carried out since 2017 in Brazil by the Atlas da Notícia (2022) initiative, developed by the Press Observatory (Observatório da Imprensa) and by the Institute for the Development of Journalism (Projor).

The most recent data from the Atlas da Notícia (2022) show that, in 2021, a total of 2968 municipalities, or 53.3%, of the 5570 municipalities in the country did not have any media at all. In 2019, there were 3280 municipalities without news outlets, which means that there was a reduction of 9.5% compared to the previous report. News deserts, as such, affect 29.3 million people—13.8% of the Brazilian population. Municipalities with one or two media outlets, called in the report "almost deserts", correspond to 26% of the municipalities, with a total population of about 32 million people. Regarding closures and news businesses, in 2021, 79 Brazilian news outlets were mapped as closed. Conversely, 642 new journalistic ventures were created between 2020 and 2021, of which 449 were native digital.

As seen in the United States (Abernathy 2020), in Brazil, news deserts are also more frequent in smaller communities with lower levels of economic activity. On average, cities in news deserts have 9800 inhabitants, with a median of 6600 people. The "almost deserts" have an average of 21,700 and a median of 14,800 inhabitants.

In comparison, 47% of municipalities across all regions of the country have at least one journalistic media outlet covering news, totaling 2602 municipalities where 182 million inhabitants live, amounting to 86% of the population.

The highest number of municipalities in news deserts is found in the north and northeast regions, where 63.1% and 62.4% of the municipalities, respectively, are in news deserts. In the north region, out of 450 municipalities, 284 do not have any local media covering local affairs (J. Botelho 2022; Correia 2022). In the southeast region, there are 881 news deserts, amounting to 52.8% of the municipalities, and in the south region, the percentage of news deserts ranges between 45% and 48% in the three states that compose the region (Fontoura 2022; Sônego 2022). Meanwhile, the midwest region holds the lowest concentration of news deserts, with 29.1% or 136 municipalities lacking local journalistic coverage (Werdemberg 2022).

In Brazil, the trend seen over the past few years has been maintained: print media continue to decline, while online media outlets have been recording a speedy growth, and new radio outlets have had a more moderate development. The main challenge to local journalism in the country has been financial sustainability, an issue aggravated by the COVID-19 pandemic (Lüdtke 2022). Communities far from large urban centers with low levels of economic activity are most susceptible to the journalism crisis. Local media have smaller audiences to help support a journalistic vehicle and few revenue sources to keep their structure running. These communities are likely to be the first to have their newspapers close during an economic crash (Abernathy 2020). In addition, dependency

on public resources tends to increase, impacting their editorial independence. Meanwhile, digital native media outlets find fewer entry barriers which can explain their rapid growth (Lüdtke 2022).

The Atlas da Notícias's mapping comprises 4670 digital media outlets, 4597 radio stations, 3214 print media outlets and 1246 television channels. From digital media, 2791 outlets are blogs or social media journalism initiatives.

In Portugal, the first systematic mapping was carried out in 2022 as part of the MediaTrust.Lab "https://mediatrust.ubi.pt (accessed on 20 December 2022)" project. The Portuguese study created one more category than the studies in Brazil and the United States. In addition to news deserts, where there are no local media outlets, and threatened cities, those with only one local newspaper or radio, the report includes a "semi-desert" category, designated for locations where there is a media outlet but it does not serve the community sufficiently; because it is a printed newspaper published less than fortnightly; or because it is a station of a radio station that is licensed to be heard in a municipality, but does not have news production made in the territory (Jerónimo et al. 2022).

The report showed that more than half of Portuguese municipalities (53.9%) are news deserts or are on the verge of becoming one (Jerónimo et al. 2022). Out of the 308 municipalities, 25.3% are in some type of news desert; that is, they have insufficient news coverage. Out of these 78 municipalities, 54 (17.5%) are in a total news desert, which means that they do not have any media outlets producing news about these territories, and 24 (7.8%) are in semi-desert status; that is, they only have less frequent or not satisfactory news coverage. It should also be noted that 88 (28.6%) are at a higher risk of becoming news deserts, as they have only one media outlet with regular news coverage.

As seen in Brazil, news deserts are more present in inland municipalities, where the population is smaller, and the economy is less dynamic. In Portugal, the districts of Lisbon, Porto, Braga and Aveiro, where the 20 most populated municipalities are located, have only three towns in news deserts, while only three coastal municipalities are in a semi-desert: Aljezur, in Faro; Albergaria-a-Velha, in Aveiro; and Óbidos, in Leiria. None are in a total news desert (Jerónimo et al. 2022).

Meanwhile, over 80% of news deserts and semi-deserts in Portugal are concentrated in the North, Center and Alentejo regions. A total of 63 of the 76 municipalities that are news deserts and semi-deserts are located in these three regions. Specifically, the districts of Beja, Bragança, Évora, Portalegre and Vila Real are those with the highest number of municipalities in some type of news desert. Among the 50 smallest municipalities in terms of population, 29 (58%) are news deserts or semi-deserts. In total, 647,422 people live in some kind of news desert, representing 6.3% of the population.

Regarding purchasing power, out of the 50 municipalities with the lowest purchasing power in the country, 22 (44%) are in deserts or semi-deserts. Comparing the purchasing power map from 2019 and the news deserts map from 2022, the correlation by region between purchasing power and the creation of local news makes itself evident (Jerónimo et al. 2022).

Looking at the type of media, there are no print newspapers covering local news regularly (meaning daily, weekly or fortnightly) in 182 Portuguese municipalities (59%). Digital media outlets are present in 151 municipalities (49%), and there are no digital media news outlets in 157 municipalities (51%). Regarding radio stations, a total of 118 municipalities (38.3%) do not have any radio stations broadcasting local news, and 17 municipalities have licensed radio stations but do not have local news coverage, which means they do not have journalists locally covering news and/or the newsroom is located in another nonbordering municipality (and are, therefore, considered semi-deserts). In 59 municipalities without frequently updated printed and digital media, radio stations are the only local news source.

### 2.3. Putting Electoral Systems in Context

Electoral systems are different in the two countries studied. In this paper, we considered the Portuguese Legislative Elections of 2022 and the Brazilian General Elections of 2022, both national in nature but with relevant differences in their processes.

In Brazil, the Electoral Code states that, in the elections relevant to this work, representatives are elected for the positions of President and Vice President of the Republic, among other posts. Held on the first Sunday of October, it takes place through universal suffrage and direct and secret ballots (Resolucao nº 23.677 2021).

In the country, voting is mandatory for people over 18 years old and optional for illiterate people, people over 70 years old and people between 16 and 18 years old. However, although it is mandatory, the secret ballot allows the voter not to vote for any candidate; the obligation is to attend the electoral college and complete the voting process. If a citizen who is obligated to vote does not attend, it is possible to justify this absence before an electoral judge up to thirty days after the election, under penalty of a monetary fine, in addition to administrative sanctions that include being prevented from applying for public office or function, receiving remuneration for public employment and obtaining a passport or identity card, among others (Electoral Code—Law nº 4.737 1965). However, justifying the vote can only occur up to three consecutive times.

The Brazilian electoral system only considers the votes that are destined in the ballot boxes for a certain candidate as valid votes. Blank and null votes are considered invalid and are not totaled by the Electoral Court. They are, however, different from abstentions, when the voter does not appear at the ballot box.

According to Silva et al. (2017), there was an idea in Brazil that the null vote meant low education of the voter, a lack of preparation to exercise their citizenship. However, an analysis based on a survey of 2482 voters in 2010 by the Latin American Public Opinion Project (LAPOP) pointed out that the null vote is used as a means of protest to show that the citizen does not believe in politics or even in the democratic system, more than being related to low levels of education or lack of interest in the country's democratic process.

The protest the policy is one of the behaviors that lead to so-called electoral indifference, which brings together blank votes, null votes and abstentions, justified or not (Ramos 2009) by:

(a) Alienation, as an absence of social responsibility and commitment.
(b) Satisfaction, as a result of the conclusion that the political reality is good.
(c) Dissatisfaction, in which the individual does not feel like an integrated part of the political sphere.
(d) Apathy, representing inaction, which may derive from different states of conscience, such as: (i) individual incapacity, when the person is incapable of understanding politics; (ii) social impotence, in which the person feels that the system does not attribute any power to the isolated citizen; (iii) indifference to the political and electoral process, when the citizen attributes greater importance to other dimensions of private life.
(e) Protest, demonstrating specific dissatisfaction with the system because one rejects: (i) the State and national political community; (ii) the representative system generally and democratic institutional arrangements; (iii) the specific political system; (iv) the conduct of government representatives (Ramos 2009, pp. 179, 183).

In 2022, in a general election marked by strong political polarization between the candidates Luiz Inácio Lula da Silva, of the Workers' Party (PT), and Jair Bolsonaro, of the Liberal Party (PL), the abstention rate in the second round of voting was, for the first time in a general election, lower than in the first round. The abstention rate in the second round stood at 20.59% of the 156 million eligible voters against 20.9% in the first round, when the highest abstention in the history of a first round was recorded (Moliterno and Reis 2022).

In Portugal, the Electoral Law for the Assembly of the Republic (commonly called the Parliament) defines that Portuguese citizens over 18 years of age have active electoral capacity, excluding those who notoriously present a limitation or serious alteration of

mental functions and those who are deprived of political rights. Suffrage in Portugal "constitutes a right and a civic duty" (Law nº 14/79 2022), which does not characterize it as a "legal duty", that is, an obligation subject to sanction (Canotilho and Moreira 2007). However, the same authors point out that "the civic duty of suffrage prevents, at a minimum, a legal understanding of "right to abstain", or from attributing electoral importance to abstention" (Canotilho and Moreira 2007, p. 672, own translation).

Between 1975, when democracy was restored in Portugal after the Estado Novo dictatorship, and 2019, the abstention rates in legislative elections rose progressively, reaching a record rate of 51.4% in that year (Pordata 2022).

Although the legislative mandate lasts four years, after the nonapproval of the State Budget for 2022, the President of the Republic, Marcelo Rebelo de Sousa, decreed the dissolution of the Assembly of the Republic and set the anticipation of the legislative elections for 30 January 2022 (Decree-Law no 91/2021 2021).

For the first time since 2005, the abstention rate dropped in the 2022 Legislative Elections, to 48.6%. Parties with parliamentary representation ran for 22 constituencies, and 230 seats in the Assembly of the Republic were up for grabs in these elections, which registered a record number of candidacies (Almeida 2022).

## 3. Materials and Methods

To analyze the correlation between abstention and news deserts, this article used data from the two countries in 2022 (Atlas da Notícia 2022; Jerónimo et al. 2022), with the numbers of abstentions provided by the National Election Commission of Portugal and by the Superior Electoral Court of Brazil. The dataset comprised information about the municipalities, their location within regions and states (for Brazil) or districts (for Portugal), their abstention rates and their nature regarding being or not being in a news desert.

Data showed abstention rates for the Portuguese legislative elections of January 2022 and for the second round of the presidential elections in Brazil in October 2022. To analyze the correlation between abstention and news deserts, this article draws on data from the two countries in 2022 regarding the municipalities in news deserts (Atlas da Notícia 2022; Jerónimo et al. 2022) and the number of abstentions provided by the National Election Commission of Portugal and by the Superior Electoral Court of Brazil.

The initial objective would be to analyze local elections' rates in both countries. However, the last municipal elections in Brazil were held in 2020, in the height of the COVID-19 pandemic, when a vaccine for the disease was not yet available. In this context, nonattendance to the polls in the first round was 23.14%, against 17.58% in 2016 and 16.41% in 2012 (Cervi and Borba 2022). Thus, since this could influence our results, we decided not to use these data. In the Portuguese case, despite taking place being in a pandemic context, the 2021 municipal elections were held in October, with about 80% of the population vaccinated. The abstention rate (46.4%) was average compared to the two previous elections: 45% in 2017 and 47.4% in 2013 (Pordata 2023). These figures show how discrepant the 2020 Brazilian election numbers were. In addition, news deserts have been systematically mapped in Brazil since 2017 and in Portugal since 2022. These are limitations of this study that aim to be addressed with future investigation.

The investigation aims to answer the following research question: is the existence of news deserts correlated to abstention rates? Our hypothesis is that higher abstention rates are found in news deserts.

First, we worked with two samples for each country: the 100 municipalities with the highest abstention rates (group A; *n* = 100) and the 100 municipalities with the lowest abstention rates (group B; *n* = 100). The news deserts variable is coded into two categories: (1) in some sort of news deserts (which include deserts and semi-deserts) and (2) non-deserts.

Data were processed using SPSS v28 (Statistical Package for Social Sciences). We tested our hypothesis with two distinct empirical strategies: First, we evaluated the differences between the two groups by counting the frequency of news deserts within these groups and

comparing abstention rates' means between the two groups by running a t-test to observe if there are significant differences between two group means, in this case, news deserts and non-deserts. Then, we ran a binary logistic regression to measure the probability of higher abstention rates being found in the news deserts category.

## 4. Results

We conducted an exploratory data analysis which allowed us to identify particularities with data distributions and variances. A Shapiro–Wilk test showed that the null hypothesis of the distribution of the samples being normal at a significance level of 0.05 was rejected for all researched samples (Brazil Group A: for news deserts, $n = 70$; $p < 0.001$; for non-deserts, $n = 30$; $p < 0.001$; Brazil Group B: for news deserts, $n = 64$; $p < 0.001$; for non-deserts, $n = 36$; $p < 0.001$; Portugal Group A: for news deserts, $n = 27$; $p = 0.009$; for non-deserts, $n = 73$; $p < 0.001$; Portugal Group B: for news deserts, $n = 25$; $p = 0.021$; for non-deserts, $n = 75$; $p < 0.001$). We also visualized data in histograms (See Figures 1–4).

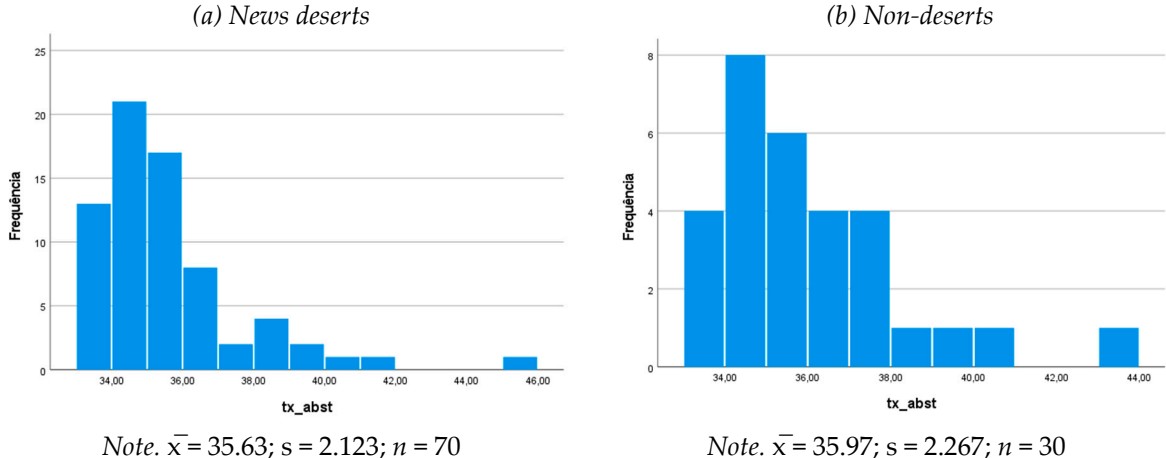

*(a) News deserts*      *(b) Non-deserts*

*Note.* $\bar{x} = 35.63$; s = 2.123; $n = 70$      *Note.* $\bar{x} = 35.97$; s = 2.267; $n = 30$

**Figure 1.** (**a**) Distribution of the continuous variable: Brazil Group A, news deserts. (**b**) Distribution of the continuous variable: Brazil Group A, non-deserts.

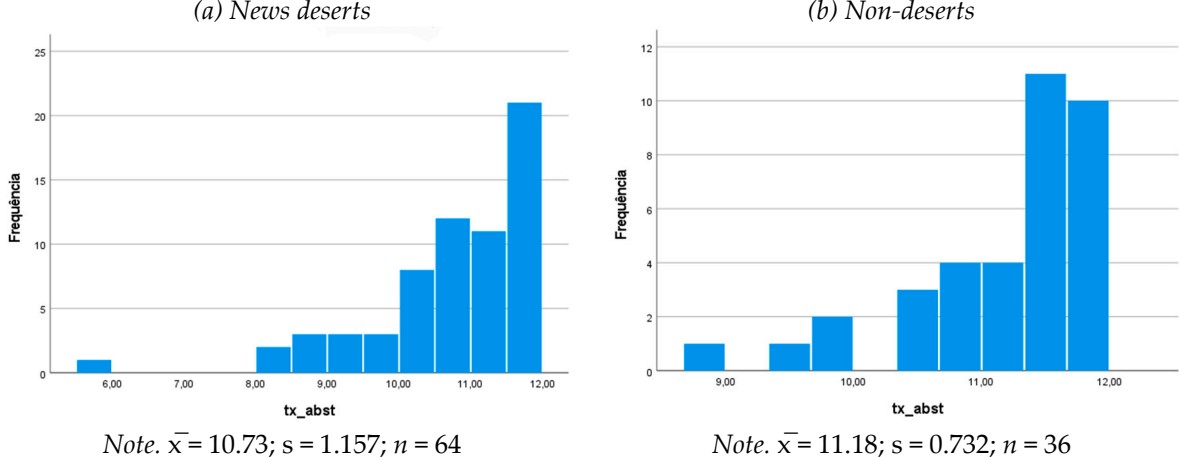

*(a) News deserts*      *(b) Non-deserts*

*Note.* $\bar{x} = 10.73$; s = 1.157; $n = 64$      *Note.* $\bar{x} = 11.18$; s = 0.732; $n = 36$

**Figure 2.** (**a**) Distribution of the continuous variable: Brazil Group B, news deserts. (**b**) Distribution of the continuous variable: Brazil Group B, non-deserts.

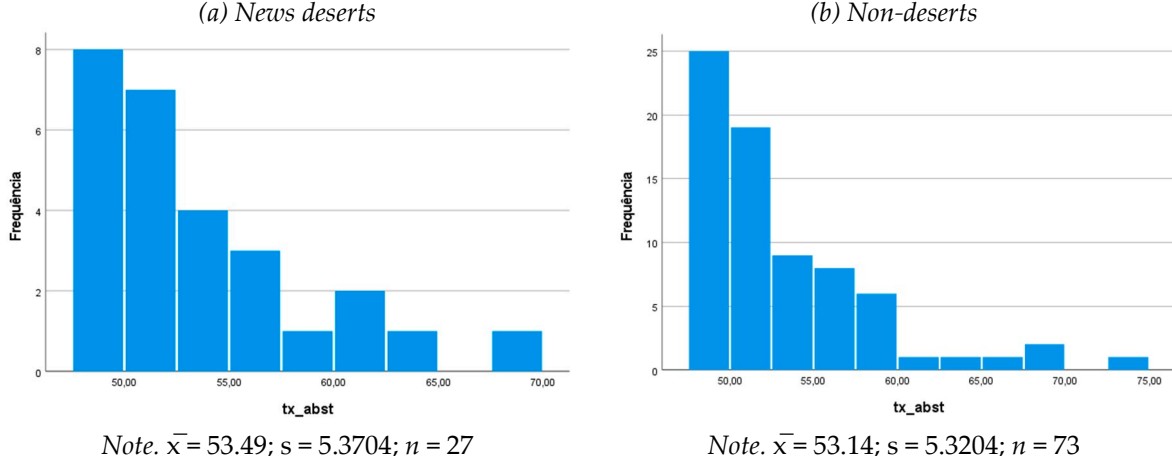

Note. x̄ = 53.49; s = 5.3704; *n* = 27        Note. x̄ = 53.14; s = 5.3204; *n* = 73

**Figure 3.** (**a**) Distribution of the continuous variable: Portugal Group A, news deserts. (**b**) Distribution of the continuous variable: Portugal Group A, non-deserts.

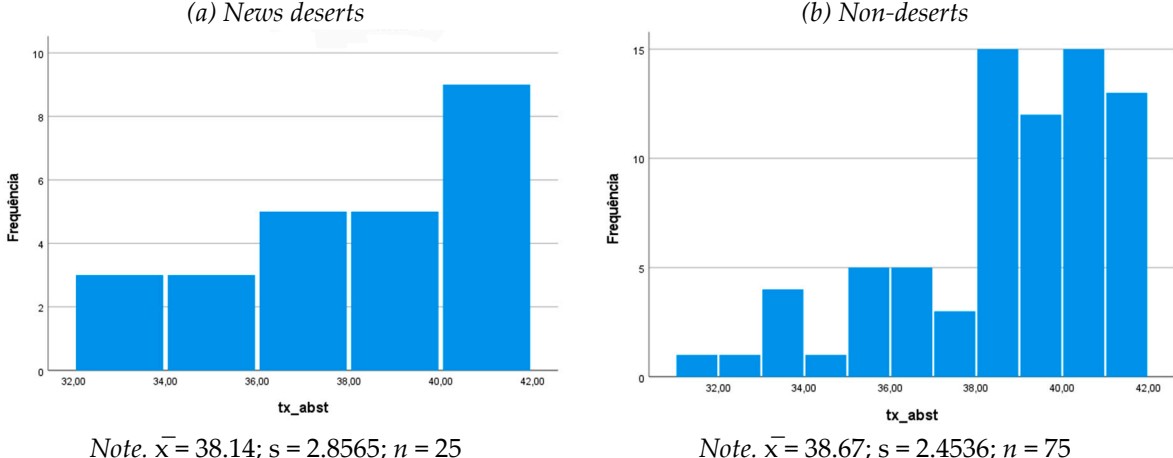

Note. x̄ = 38.14; s = 2.8565; *n* = 25        Note. x̄ = 38.67; s = 2.4536; *n* = 75

**Figure 4.** (**a**) Distribution of the continuous variable: Portugal Group B, news deserts. (**b**) Distribution of the continuous variable: Portugal Group B, non-deserts.

Levene's test based on the mean also showed that for Portugal, in both groups A and B, the null hypothesis of news deserts and non-deserts having equal variances was not rejected at a significance level of 0.05 ($p$(A) = 0.759; $p$(B) = 0.111). For Brazil's sample, however, in group A, the null hypothesis was not rejected ($p$ = 0.502), while in group B, for the variance between deserts and non-deserts, the null hypothesis was rejected ($p$ = 0.025).

We then investigated the frequency of news deserts among groups A and B. For Brazil's sample, in group A, 70 were in news deserts, while in group B, a similar number of 64 were also in news deserts. For Portugal, the same tendency was found. In group A, there were 27 news deserts, while in group B, 25 were news deserts (see Table 1).

**Table 1.** Frequency of news deserts among groups A and B in Brazil and Portugal.

|  |  | News Deserts | Non-Deserts | Total |
|---|---|---|---|---|
| Brazil | Group A | 70 | 30 | 100 |
|  | Group B | 64 | 36 | 100 |
| Portugal | Group A | 27 | 73 | 100 |
|  | Group B | 25 | 75 | 100 |

This initial analysis suggested no clear tendency of the data. We then looked at the means in these two samples observing news deserts and non-deserts. In Brazil, the

abstention rates' means were slightly higher in non-deserts, considering both groups A and B, while in Portugal, the abstention rates' means were slightly higher in news deserts for group A and slightly lower in non-deserts for group B (see Table 2).

**Table 2.** Abstention rates means in news deserts and non-deserts in Brazil and Portugal.

|  |  | Group A | Group B |
|---|---|---|---|
| Brazil | News deserts | 35.63 | 10.73 |
|  | Non-deserts | 35.97 | 11.18 |
| Portugal | News deserts | 53.49 | 38.14 |
|  | Non-deserts | 53.14 | 38.67 |

To compare if there were significant differences between the means, an independent-samples t-test found that in both groups A and B of Portugal, we cannot reject the null hypothesis that the average abstention rate is equal between news deserts and non-deserts (for group A, $n = 100$; $p = 0.768$; t = 0.297; for group B, $n = 100$; $p = 0.409$; t = −0.835) at a significance level of 0.05.

For Brazil's group A, the same trend was found ($n = 100$; $p = 0.494$; t = −0.689; $\alpha = 0.05$); thus, we cannot also reject the null hypothesis. However, when it comes to group B in Brazil, the test found significance ($n = 100$; $p = 0.019$; $\alpha = 0.05$); thus, we reject the hypothesis that the average abstention rate is equal between the two categories.

To test the correlation between the two variables investigated, we decided to use the binary logistic regression since the investigation in this paper is focused on a model in which the dependent variable is categorical and binary (news deserts and non-deserts) and the data are not normally distributed. We investigated if this variable is correlated to the abstention rates. Other correlation tests such as Pearson's r, Spearman's rho or point-biserial correlation could not be made as many of the assumptions of these tests were not met (Tranmer and Elliot 2008).

Drawing on a statistical model to investigate the relationship between variables, we investigated what the odds are of news deserts having higher abstention rates.

Beginning with Brazil's group A sample, the Hosmer–Lemeshow test was found to be not significant ($p = 0.609$), indicating an acceptable fit of the model. The sensitivity for the model is that 100% of the cases fall into the news deserts category for both of Brazil's samples. The overall classification accuracy is 70%. Taking a look at the regression table, we found that the test was not statistically significant ($p = 0.479$; B = 0.069). For Brazil's group B, the Hosmer–Lemeshow test was also found to be not significant ($p = 0.685$), but in the regression table, the test was found to be significant at a 0.05 level ($p = 0.043$) and the predicted logit was 0.525, showing that, from below high levels of abstention, the predicted probability of it falling in the news desert category increases.

For Portugal's group A, the Hosmer–Lemeshow test also indicated an acceptable fit of the model ($p = 0.441$). However, again, the test was not statistically significant ($p = 0.764$; B = −0.013). For Portugal's group B, again, the Hosmer–Lemeshow test was not significant ($p = 0.875$). The regression test was not significant ($p = 0.367$; B = 0.079).

When testing regression coefficients for statistical significance, we can see that in both Portugal's samples and in Brazil's group A, the abstention rate variable was not a significant predictor in the model. Taking a look at both tests, we found that in Portugal, we cannot say that the variables were correlated, while in Brazil, news deserts were correlated only in the lower-abstention-rates group.

## 5. Conclusions and Discussion

The inland territories of both Brazil and Portugal tend to have smaller populations and less dynamic economies. One consequence is that there are fewer local media outlets. It is clear that news deserts are prevalent in the less developed regions of the country, where there is a shortage of funding to support journalistic activities—something which is the

same in both countries. As explained earlier, as smaller territories retain fewer economic resources, the traditional business model of small-scale newsrooms is becoming less and less sustainable as they face one economic crisis after another.

These factors, combined with a higher degree of isolation, difficulties in transportation in small territories, less integration with the central governments and less information being produced and disseminated, contribute to creating a more fragile public sphere and lower interest in civic matters (Carvalho 2017) and opening these territories to threats such as the spread of hate speech and disinformation (Jerónimo and Esparza 2022).

The lack of statistical significance for the binary logistic regressions in the highest-abstention-rates groups (group A) both in Brazil and Portugal and the lack of difference between groups in the t-test show that we cannot state that the inexistence of media outlets influences the mobilization of the electorate to vote. Thus, answering our research question, it was not possible to assess the correlation between the (in)existence of news deserts and higher or lower abstention rates in national elections. Our findings indicate that the presence or lack of local news is not the main factor mobilizing people to their polling places, indicating that other factors have more influence on each person's decision to vote both in Brazil and in Portugal, despite differences in the mandatory nature of voting.

In Brazil, part of the population may not vote because they are unable to travel to the polling station, for example. In Portugal, however, this factor may have some relevance, but the "no votes" behavior could also be influenced by the lack of interest in the political process. Therefore, further studies need to test the correlation with other variables so these dynamics can be better understood.

Some factors affecting this study need to be addressed. The first point is the reports on which this comparison was based, the "Atlas da Notícia" and the "News Deserts Europe 2022: Portugal Report", which use different methodologies to gather information about news outlets and to map them. In Portugal, media are registered in the ERC (the regulatory entity), which lists the official regulated media outlets within the country. In Brazil, on the other hand, there is no media regulation entity that performs this work, and so the Atlas da Notícia considers outlets found through their own research and by their regional partners. Since the methodologies are different, it must be considered that it may affect the results. The criteria to define what is a newspaper or a media outlet may be different, but we argue that it is possible to make a comparison since both countries carried out systematic mapping based on a defined methodology, as has been performed in a few countries around the world. It is also important to highlight the differences in the electoral processes of Brazil and Portugal examined in this paper: in the first, citizens vote in national elections to choose representatives for executive and legislative mandates, with mandatory voting; in the latter, citizens vote to choose parliamentarians, with optional voting. A direct comparison between abstentions cannot be made; however, the relationship in each country between voting abstentions and news deserts is discussed here.

It is noteworthy that the analysis draws on data from abstentions in national and not local elections. As stated in the methodology, the 2020 municipal elections in Brazil were hampered due to the pandemic that affected abstention rates (Cervi and Borba 2022). However, local journalism also reports on national elections, as they impact citizens in all localities. Further analyses of a possible correlation between news deserts and abstentions in local elections should be carried out in 2025, drawing on data from 2024's (Brazil) and 2025's elections (Portugal).

Despite the difference in the countries' situations, local media play an important role in making people aware of the voting calendar and the process, in the scrutiny of the candidates and their proposals, and in providing a floor to debate the social problems that communities face (Abernathy 2022). However, the media in general, and their role as a watchdog of power, also play an important role in building a negative image of politics, showing corruption, internal disputes and scandals, which may contribute to the public's disinterest in the electoral process (Muñoz-Alonso Ledo 2004). In either way, as this study found, the existence of local media does not mean that people feel mobilized enough to

leave their houses and go to their polling places to vote. Thus, although it is seen that news deserts are more prevalent in regions with a smaller population and less economic activity (Abernathy 2018), it is not possible to point out that the lack of media correlates with the lack of polling participation.

People obtain information in these news deserts somehow. They do not obtain information through journalism, but they obtain information in other ways. How are they informed? Social media may have become an important provider of local information, as people gather in groups or profiles which are usually geographically identified with their territories (Collier and Graham 2022). It is essential to develop further studies in these news deserts to understand how people obtain information about their territories and to what extent the people in them are exposed to, for instance, disinformation. In addition, it is also convenient to discuss the concept of news deserts, as we understand—in line with what Usher (2023) advocates—that other types of media (alternative, community, etc.) have not been considered in this kind of study. Additionally, it is extremely relevant to understand which factors determine citizens' voting behavior. Studying how people obtain information and how this influences their voting behavior is fundamental to understanding how democracies work where there is no reliable information being broadcast to the public about them.

**Author Contributions:** Conceptualization, G.R., L.T. and P.J.; methodology, G.R. and L.T.; formal analysis, L.T.; investigation, G.R. and L.T.; data curation, G.R. and L.T.; writing—original draft preparation, G.R. and L.T.; writing—review and editing, P.J.; visualization, P.J.; supervision, P.J.; project administration, P.J.; funding acquisition, P.J. All authors have read and agreed to the published version of the manuscript.

**Funding:** This work was supported by the Fundação para a Ciência e a Tecnologia, Portugal, under grant reference PTDC/COM-JOR/3866/2020.

**Institutional Review Board Statement:** Not applicable.

**Informed Consent Statement:** Not applicable.

**Conflicts of Interest:** The authors declare no conflict of interest.

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
