# Peer review of "No Media, No Voters? The Relationship between News Deserts and Voting Abstention"

_socsci, doi:10.3390/socsci12060345_

Round 1

Reviewer 1 Report

1. The text provides insightful information on the relationship between the presence of media outlets and the mobilization of citizens to vote in two contexts (Brazil and Portugal) affected by news deserts at the local level. This problem has been widely examined by the literature in recent years, but the article offers an original approach that explores the democratic consequences of the lack of local news provision. It is noted the difficult digital transition of local media as a reason of the disappearance of media outlets, but the authors acknowledge that “desert” refers not only to the absence of local media, but also to the difficulty in access those media.

2. The article has a scientific structure. A clear research question and a valid hypothesis are provided, using statistical tests to respond both. Maybe it would be helpful to come back to this information (research question and hypothesis) in a more formal way in the Conclusion. This final section should be named as “Conclusion and Discussion” since it fosters a debate on how to study the phenomenon. The current text shows the aim of the authors and, therefore, what is the gap addressed.

3. The list of references is up-to-date and suitable for the present study.

4. This paper allows that one can clearly learn a lot about potential implications of the absence of local media in democracy. In this sense, the authors found that it was not possible to assess a proper correlation. In Portugal, there was no correlation; meanwhile, a few was detected in the lower abstention rates groups in Brazil.

5. Finally, some limitations of the study are addressed and the current version reflects upon the fact that less developed regions are the ones with a weaker news provision. However, it is missed a wide discussion on the different nature of the elections analyzed. The reports on local media outlets, in which the comparison was based, are discussed, but maybe it could be more relevant to examine regional or local elections (for instance, “eleições autárquicas” in Portugal). Besides that, general elections (Brazil) and legislative elections (Portugal) have different implications. Why have elections not been studied at the same level? This would enhance the value of the comparison.

6. In short, this paper is an interesting contribution for the communication field, particularly aligned with prior scholarship on the state and scope of local news. Some suggestions are here provided, but I do think that the article is publishable in present form.

Reviewer 2 Report

Relevance and research gap

A more exhaustive state of the question on the studies that have related the political behavior of the citizenry and the absence/presence of media is necessary to reflect the scientific gap that the work intends to complete and to demonstrate its interest.

Scope and aims

I consider that the article has the wrong approach. In fact, the research question should be: is the existence of 330 news deserts correlated to abstention rates in national elections? This is the main problem of the research. The analysis of data referring to legislative or presidencial elections is not sufficiently justified. How can the existence of a local media have an impact on national elections, which are widely followed by national media? In fact, the paper alludes to the fact that citizens do receive national information (“the citizens have access to trustworthy information about what is going on in their countries or in the world, but not about their local communities”, p. 3). In what sense then could it have an impact if there is or is not a local media? It is recommended to refocus the study with a sample of municipalities (considered as information deserts and non-deserts) and municipal elections. Obviously, this cannot be done with the entire population of the entire country, but it can be test with a sample of municipalities. SIMILAR CASE STUDIES?

Methodology

There are risks in categorizing municipalities as deserts or not, as authors such as Usher have already questioned, because the fact that there is no local media does not mean that political information does not arrive through other channels, but neither does the existence of a local media guarantee that it includes political information. Trying to establish relationships in this sense would require a more in-depth examination of the media and determine the type of  coverage they offer. The methodology used, based solely on the dichotomy of the existence or non-existence of media from a structural point of view, does not lead to a clear answer to the research question. As a result, the findings are rough, with little nuance and lead to generalizations that may not correspond to reality.

This text of the conclusions should be part of the methodology, where it should be made clear which methodology is used by each country's database to measure news deserts: “the reports in which this comparison was based, the “Atlas da Notícia” and the “News Deserts Europe 2022: Portugal Report”, which use different methodology to gather information about news outlets and to map them. In Portugal, media are registered in ERC (the regulatory entity), which lists the official, regulated media outlets within the country. In Brazil, on the other hand, there is no media regulation entity that does this work, and so the Atlas da Notícia considers outlets found through their own research and by their regional partners”.

In addition, the methodology used by both news desert databases (Brasil and Portugal) should be pointed out to clarify what is included and what is not. In Brasil “blogs e veículos de redes sociais” are not only included, but represent 59% of digital media (Atlas da notícia, 2022). In Portugal, by contrast, this type of media is not part of the sample. They do not register the same types of media, therefore, the information deserts of one country cannot be equated with those of the other and the results could be biased.

The electoral systems are also different: the set of voters is different, because in Brazil there is a set of citizens for whom voting is optional, something that does not happen in Portugal.

Conclusions

The conclusions require more depth and discussion with other works that adopt the theoretical framework of news deserts. They include paragraphs that are not based on evidence or bilbliography and cannot be concluded with the study conducted. For example: “Despite the difference in the countries’ situations, local media plays an important role in making people aware of the voting calendar and the process, in the scrutiny of the candidates and their proposals, and in providing a floor to debate social problems communities face. However, the media in general, and its role as a watchdog of power, also plays an important role in building a negative image of politics, showing corruption, internal disputes and scandals, which may contribute to the public’s disinterest in the electoral process.

References

It is suggested to resort to more scientific literature and not to journalistic or informative texts, especially on the case of Brazil.

Reviewer 3 Report

I want to thank the author for the opportunity to read his work. The article “No Media, No Voters? The Relationship between “News Deserts” and Voting Abstention” represents a valuable contribution to various disciplines, from political sciences, communication studies, sociology, and international relations, among other related disciplines. The following comments are driven to increase the overall quality, clarify some points and offer a constructive review.

1. The article aims to forecast and discuss an interesting issue: the relationship between information and voting (the acting, not the electoral preferences). This such it must be said that it’s an interesting proposal considering the vast majority of available literature focus on these links to explain electoral behavior or misinformation.

2. The literature on local news/local journalism and news deserts are updated, and the links with the given context are also carefully done. It’s a strengthening point of the article the presence of global south scholars to explain and debate the context and the given phenomena (not getting it limited to the contextualization).

3. The differences in the studied political systems are also explained. Although it is a minor adjustment, it should be explicated as a limitation of the research.

4. The findings and discussion section is clearly written. Nevertheless, it must incorporate the theoretical framework mobilized previously to discuss it effectively. In this sense, authors’ ideas on news deserts and so on should be explicitly mentioned (it is between the lines).

These minor reviews can make the article more accurate to a broader audience, and we believe can collaborate with it in general. Despite it, I believe it addresses an important issue and has an interesting approach.

Round 2

Reviewer 2 Report

I appreciate the authors' willingness to address the recommendations made. I would like to point out that what I raised was not a minor revision; on the contrary, I considered that the focus of the article was not accurate: the study on national elections. This has not changed.

However, given that the method used and the analysis are adequate and the justification of interest has been improved, the work can be published.

Further recommendations

In order not to approach the study of media deserts from a limiting point of view, it would be interesting in the future study of local elections not only to take into account the (in) existence of local media but also the influence of the availability of political information on voter turnout, as Baekgaard et al (2014) did in their study.

Regarding the Brasilian Atlas da Notícia, although the data correspond to work carried out by the researchers, it is recommended that they be published in scientific publications, not only in blogs.